# Arabidopsis 3β-Hydroxysteroid Dehydrogenases/C4-Decarboxylases Are Essential for the Pollen and Embryonic Development

**DOI:** 10.3390/ijms242115565

**Published:** 2023-10-25

**Authors:** Jiawen Pan, Weifeng Li, Binzhao Chen, Linchuan Liu, Jianjun Zhang, Jianming Li

**Affiliations:** 1State Key Laboratory for Conservation and Utilization of Subtropical Agro-Bioresources, College of Forestry and Landscape Architecture, South China Agricultural University, Guangzhou 510642, China; panjw@stu.scau.edu.cn (J.P.); liweifwng@stu.scau.edu.cn (W.L.); cbz@stu.scau.edu.cn (B.C.); lcliu@scau.edu.cn (L.L.); 2Department of Biology, Hong Kong Baptist University, Kowloon, Hong Kong

**Keywords:** sterol biosynthesis, 3β-hydroxysteroid dehydrogenase/C4-decarboxylase, C4-demethylation, pollen development, embryogenesis

## Abstract

The biosynthesis of C_27–29_ sterols from their C_30_ precursor squalene involves C24-alkylation and the removal of three methyl groups, including two at the C4 position. The two C4 demethylation reactions require a bifunctional enzyme known as 3β-hydroxysteroid dehydrogenase/C4-decarboxylase (3βHSD/D), which removes an oxidized methyl (carboxylic) group at C4 while simultaneously catalyzing the 3β-hydroxyl→3-keto oxidation. Its loss-of-function mutations cause ergosterol-dependent growth in yeast and congenital hemidysplasia with ichthyosiform erythroderma and limb defect (CHILD) syndrome in humans. Although plant 3βHSD/D enzymes were well studied enzymatically, their developmental functions remain unknown. Here we employed a CRISPR/Cas9-based genome-editing approach to generate knockout mutants for two Arabidopsis *3βHSD/D* genes, *HSD1* and *HSD2*, and discovered the male gametophytic lethality for the *hsd1 hsd2* double mutation. Pollen-specific expression of *HSD2* in the heterozygous *hsd1 hsd2/+* mutant not only rescued the pollen lethality but also revealed the critical roles of the two *HSD* genes in embryogenesis. Our study thus demonstrated the essential functions of the two Arabidopsis *3βHSD/D* genes in male gametogenesis and embryogenesis.

## 1. Introduction

Sterols are isoprenoid-derived lipids that have important physiological activities for all eukaryotic organisms, especially for plants that synthesize a wide variety of terpenoids including carotenoids, dolichols, tocopherols, chlorophylls, and squalene [1]. Sterols are significant components of eukaryotic cell membranes, which not only regulate the permeability and fluidity of membranes, the activity of membrane binding proteins, and the integrity of various cellular membranes but also participate in many membrane-related metabolic processes and vesicle transport [2,3]. In addition, sterols are biosynthetic precursors of various steroid hormones, including mammalian androgens and estrogens, insect ecdysteroids, fungal antheridiol and oogoniol, and plant brassinosteroids (BRs) [4]. Some sterols and sterol biosynthesis intermediates also function as signaling molecules to regulate transcription and other cellular activities [5].

In the sterol biosynthesis pathways of plants, animals, and fungi, sterols only become functional after the sequential removal of two methyl groups at the C4 position and one methyl group at the C14 position. Therefore, the C4 demethylation of sterol intermediates is a critical step in sterol biosynthesis. This step is catalyzed by the sterol C4 demethylase complex (SC4DM), which is highly conserved from fungi to plants to humans [6,7,8]. SC4DM is composed of sterol C4 methyl oxidase (SMO), 3βHSD/D, and 3-keto steroid reductase (3KSR). These individual enzymes are connected by ergosterol biosynthesis protein 28 (ERG28), which acts as a scaffold to anchor the SC4DM components to the endoplasmic reticulum (ER) membrane [9,10]. In animals and yeast, the two C4 methyl groups are sequentially removed early in the sterol biosynthetic pathway right after the C14-demethylation step. In higher plants, the first C4 methyl group is removed early in the cyclopropyl sterol precursor, and the second C4 methyl group is removed only after the C14 demethylation. Therefore, the demethylation sequence in animals and yeast is C14, C4, and C4, while in plants, it is C4, C14, and C4. The separation of the two C4-demethylation steps by the C14-demethylation might allow plants to produce unique 4-methylated sterols important for plant terrestrialization and plant stress tolerance [7,11]. Despite the difference in the order of the three demethylation reactions, the enzymatic reactions are identical. The C4 demethylation is initiated with SMO, which oxidizes a C4-methyl to a carboxyl group. Then, the 3β-hydroxyl group on the A-ring is oxidized to a ketone group by 3βHSD/D, which also catalyzes the C4 decarboxylation. Finally, the 3-keto group is reduced back to the 3β hydroxyl group by 3KSR [9,12].

Many studies have revealed the importance of the C4 demethylation reactions in various organisms. In animals, defects in the C4 demethylation process may affect the synthesis of all types of steroid hormones and lead to embryonic lethality [13]. Similarly, yeast mutants defective in the C4 demethylation process are lethal and are therefore absolutely relying on exogenously supplied ergosterols for growth [14]. In Arabidopsis, loss-of-function mutations in SMO, which are encoded by two distinct gene families, *SMO1* (with three members known as *SMO1-1*, *SMO1-2*, and *SMO1-3*) and *SMO2* (containing two members, *SMO2-1* and *SMO2-2*), resulted in stunted growth in shoots and roots (weak alleles) and embryo lethality (strong alleles) [15,16]. However, it remains unknown whether the 3βHSD/Ds are essential for plant development despite the fact that they were the first components of the plant SC4DM complex to be characterized. Rahier et al. [17] identified two Arabidopsis proteins, At3βHSD/D1 (At1g47290) and At3βHSD/D2 (At2g26260), which exhibit high sequence similarity to the yeast ERG26 and mammalian 3βHSDs catalyzing the 3β-OH to 3-oxo conversion. Detailed biochemical studies coupled with molecular modeling not only revealed that these two enzymes only accept sterol substrates with 3β-hydroxyl and C4 carboxyl groups but also identified their key catalytic residues [17,18]. Virus-induced gene silencing of a tobacco (*Nicotiana benthamiana*) homolog of the two Arabidopsis *3βHSD/D* genes resulted in reduced leaf growth [17]. Interestingly, although the simultaneous elimination of the two Arabidopsis genes had little impact on plant growth, their overexpression led to observable growth defects such as shorter inflorescent internodes with clustered siliques and unequal leaf expansion [19]. These relatively weak phenotypes were quite different than what was observed with the Arabidopsis mutants lacking the two SMOs. This could be explained by the presence of two additional homologs of 3βHSD/D in Arabidopsis [19] or by the possibility of the reported T-DNA double insertional mutant of *At3βHSD/D1* and *At3βHSD/D2* not being a true double knockout mutant due to the intron location of T-DNA in the analyzed mutant of *At3βHSD/D2*.

To investigate the physiological functions of the two reported Arabidopsis At3βHSD/D enzymes (renamed here HSD1 (At1g47290) and HSD2 (At2g26260) for simplicity), we employed a CRISPR/Cas9-based genome editing approach to create a true double knockout mutant. We discovered that while single mutants of *HSD1* and *HSD2* had no detectable effect on plant growth, we were unable to obtain an *hsd1 hsd2* double mutant. Reciprocal genetic crosses of the two heterozygous double mutants, *hsd1 hsd2/+* or *hsd1/+ hsd2*, with the wild-type control revealed lethality of the double mutant pollens, which could be fully rescued via pollen-specific expression of the *HSD2* gene driven by a widely used pollen-specific promoter *pLAT52* [20]. Our phenotypic analyses of the resulting *pLAT52::HSD2-FLAG hsd1 hsd2/+* transgenic mutants also revealed the essential function of the two HSDs in embryogenesis. Taken together, our study concluded that HSD1 and HSD2 are essential for the plant reproductive development in Arabidopsis.

## 2. Results

### 2.1. Generation of hsd1 and hsd2 Mutants via CRISPR/Cas9-Based Genome Editing

An earlier study that investigated the physiological functions of HSD1 and HSD2 used an *hsd1 hsd2* double mutant carrying a T-DNA insertion in each gene but observed no phenotypic change compared to its corresponding wild-type control [19]. However, we suspected that the reported *hsd1 hsd2* double mutant might not be a true double knockout mutant because one of the T-DNAs was actually inserted into the seventh intron of the *HSD2* gene [19]. In order to analyze the biological importance of these two *HSD* genes, we decided to create a true *hsd1 hsd2* double knockout mutant using the CRISPR/Cas9-mediated genome editing approach. We first generated a *CRISPR/Cas9-HSD1/2* transgene carrying four target DNA segments, designated as *sgRNA-DT1*, *sgRNA-DT2*, *sgRNA-DT3*, and *sgRNA-DT4*, with the first two targeting *HSD1* and the last two for *HSD2* (Figure 1a) and transformed this *CRISPR/Cas9-HSD1/2* transgene into the wild-type Arabidopsis plants. The T1 transgenic lines were screened for the presence of an antibiotic-resistant gene and their self-pollinated T2 offspring were examined by PCR for *CRISPR/Cas9* transgene-free plants carrying mutations of the two target *HSD* genes. The candidate *hsd1/hsd2* mutants were subsequently verified by sequencing of PCR-amplified *HSD1/HSD2* genomic fragments. Sequencing analysis revealed a single adenine nucleotide insertion at position 423 bp downstream of the initiation codon of the *HSD1* mRNA in the *hsd1* mutant line. This insertion is in the third exon, likely causing a frameshift and premature translational termination of the *HSD1* mRNA (Figure 1b,c). Our DNA sequence analysis found that the *hsd2* mutant carries a 46-bp deletion in the *HSD2* gene, with the missing nucleotides located in the second exon. Such a deletion also causes a frameshift mutation and premature translational termination of the *HSD2* transcript (Figure 1b,c).

PCR-based genotyping of the T3 generation identified *hsd1* and *hsd2* single mutants, transheterozygous double mutants (*hsd1/+ hsd2/+*), and two heterozygous double mutants (*hsd1 hsd2/+* and *hsd1/+ hsd2*). However, no homozygous *hsd1 hsd2* mutant was obtained. Phenotypic analyses of these mutants revealed no growth defects in *hsd1*, *hsd2*, *hsd1 hsd2/+*, or *hsd1/+ hsd2* mutant at the seedling and maturation stages, most likely due to their functional redundancy (Figure 2a,b). These results were consistent with the earlier report that mutations in the Arabidopsis *HSD1* and/or *HSD2* genes had little impact on vegetative growth [19]. However, our failure to obtain an *hsd1 hsd2* double homozygous mutant suggested that the two *HSD* genes have essential redundant functions in gametogenesis and/or embryogenesis.

### 2.2. The hsd1 hsd2 Double Mutation Impairs Development of Male Gametophytes

In order to determine the genetic basis for our failed attempts to obtain a single *hsd1 hsd2* double mutant, we analyzed the segregation ratio of the self-fertilized progeny of *hsd1/+ hsd2* and *hsd1 hsd2/+* heterozygous double mutants. As shown in Table 1, both mutants exhibited a segregation ratio of ~1:1 of single mutant:heterozygous double mutant (93:85 for *hsd1/+ hsd2* and 103:97 for *hsd1 hsd2/+*), which deviated from the normal Mendelian segregation ratio of 1:2:1 of single:heterozygous double:homozygous double mutants. These results suggest that the *hsd1 hsd2* double mutation likely causes gametophytic defect(s). To determine whether the observed gametophytic defect of the *hsd1 hsd2* double mutation was associated with the male or female gametes, or both, we made reciprocal genetic crosses of the *hsd1/+ hsd2* and *hsd1 hsd2/+* mutants with the wild-type plants. When the two mutants were used as the female recipients, the resulting F1 offspring had a ratio of double heterozygous vs. single heterozygous mutants very close to 1 (1.04 for the WT ♂ × *hsd1/+ hsd2* ♀ cross and 0.96 for the WT ♂ × *hsd1 hsd2/+* ♀ cross) (Table 2), indicating that the female gametes of the two mutants were transmitted normally. By contrast, when the wild-type plants were pollinated with the pollens of the two heterozygous double mutants, the F1 offspring produced only single heterozygous mutants but no double heterozygous mutant, revealing the male gametophytic defect of the *hsd1 hsd2* pollens. In comparison, the genetic crosses using pollens of the single *hsd1* or *hsd2* mutant to pollinate the wild-type flowers produced all single heterozygous plants, indicating neither *hsd1* nor *hsd2* male gametes had a gametophytic defect. Together, these results revealed a serious male gametophytic defect of the *hsd1 hsd2* double mutation.

### 2.3. The hsd1 hsd2 Double Mutation in Heterozygosity Does Not Affect Flower Formation

The reciprocal cross experiments of the two heterozygous double mutants, *hsd1/+ hsd2* and *hsd1 hsd2/+*, with the wild-type plants revealed that the *hsd1 hsd2* double mutation had severe male gametophyte defect(s) (Table 2). Therefore, we were interested in determining whether or not the *hsd1* or/and *hsd2* mutation had any detectable impact on the flower development. As shown in Figure 3, the flowers of all of the analyzed mutants were similar to those of wild-type plants in terms of the overall flower size, the number of individual floral organs, and the number of flowers per plant. Careful inspection of the flowers of the mutant plants detected no obvious abnormality of various floral organs. Furthermore, their anthers were able to naturally release pollens, and the stamens extended normally to allow the released pollens to come into contact with the stigma, leading to normal silique and seed development.

### 2.4. The hsd1 hsd2 Double Mutation Causes the Male Pollen Lethality

The male gametophytic dysfunction is often caused by defective pollen development. To examine whether or not the pollen viability of the *hsd1/+ hsd2* and *hsd1 hsd2/+* mutants was affected, we stained mature anthers from the *hsd1*, *hsd2*, *hsd1/+ hsd2*, and *hsd1 hsd2/+* mutants and their wild-type control with Alexander’s stain, which colors aborted pollen grains “blue-green” while staining viable pollen grains “magenta-red” [21]. As shown in Figure 4, the pollens from the wild-type and the two single mutants were stained purple-red, whereas the anthers of the *hsd1/+ hsd2* and *hsd1 hsd2/+* mutants contained both normal purple-red-colored and deformed blue-green stained pollen grains (Figure 4). This dye-based pollen viability test strongly suggested that the *hsd1 hsd2* double mutation likely caused pollen lethality, explaining the failure to obtain the *hsd2*-containing F1 offspring and the *hsd1*-carrying F1 offspring from the *hsd1 hsd2/+* ♂ × wild-type ♀ and *hsd1/+ hsd2* ♂ × wild-type ♀ crosses, respectively.

### 2.5. The Male Gametophytic Defect of the hsd1 hsd2/+ Mutant Was Complemented by the Pollen-Specific Expression of the HSD2 Gene

To verify that the observed pollen defect in the *hsd1 hsd2/+* mutant was indeed caused by the *hsd1 hsd2* double mutation rather than by an unknown CRISPR/Cas9-created off-target mutation, we created a *pLAT52::HSD2-FLAG* transgene, containing the *pLAT52* promoter, one of the best studied pollen-specific promoter from the tomato gene *LAT52* [20], the entire coding sequence of the Arabidopsis *HSD2* gene, and the coding sequence of the widely used FLAG epitope tag, and transformed the resulting transgene into the *hsd1 hsd2/+* mutant. PCR-based genotyping of the resulting T1 transgenic plants identified several *pLAT52::HSD2-FLAG hsd1 hsd2/+* transgenic mutants. Their anthers were collected and subsequently stained with Alexander’s dye. As shown in Figure 5, almost all pollens of the analyzed transgenic lines were stained “magenta-red”, indicating that the pollen lethality was fully rescued by the *pLAT52::HSD2-FLAG* transgene.

### 2.6. The hsd1 hsd2 Double Mutation Likely Caused the Embryo Lethality

Our PCR-based screening of the *pLAT52::HSD2-FLAG* transgenic lines failed to identify a single *pLAT52::HSD2-FLAG hsd1 hsd2* transgenic double mutant, despite that the introduced transgene was able to successfully rescue the pollen lethality phenotype (Figure 5). We suspected that the *hsd1 hsd2* double mutation might also cause embryo lethality, a phenotype that was previously reported in Arabidopsis mutants lacking the two SMOs [15,16], which act upstream of the 3βHSD/D enzyme in demethylating the two C4 methyl groups. Indeed, when we examined green mature siliques of several independent *pLAT52::HSD2-FLAG hsd1 hsd2/+* transgenic lines, and discovered that while most seeds were healthy and green, approximately 25% of the seeds were small, shrunken, and brown-colored (Figure 6a,b). After maturation, almost all of the normal-looking seeds were germinated, whereas those small/shrunken seeds were not. Together, these experiments strongly suggested that the *hsd1 hsd2* double mutation likely caused embryo lethality, and the pollen-specific *pLAT52::HSD2-FLAG* transgene was not able to complement the seed/embryo defects.

## 3. Discussion

### 3.1. The CRISPR/Cas9-Genome Editing Failed to Create a True hsd1 hsd2 Double Mutant

The Arabidopsis genome contains at least two genes encoding bifunctional 3βHSD/D enzymes, HSD1(At1g47290) and HSD2(At2g26260), which catalyze two separate C4 demethylation reactions in phytosterol biosynthesis [17]. It was claimed that they were the first plant hydroxysteroid dehydrogenases to be molecularly characterized and biochemically investigated [17], yet their physiological functions remain unknown. An earlier study using a presumed Arabidopsis double mutant carrying T-DNA insertions in these two genes did not detect any development or growth defect [19]. This could be caused by the functional redundancy between the two 3βHSD/D enzymes and their homologs. An earlier sequence analysis revealed two additional Arabidopsis proteins exhibiting sequence similarity with the two Arabidopsis HSD enzymes: At2g43420 and At2g33630 [19]. At2g43420 is also known as RETICULON 20 (RTN20), a member of the Arabidopsis RTN family that was thought to be involved in the formation of the ER tubules [22] and was recently implicated in sterol/lipid biosynthesis [23]. At2g33630 is also known as the Arabidopsis member of the short-chain alcohol dehydrogenase/reductase (SRD) subfamily 42E [24]. Importantly, a recent study revealed the role of the human SDR42E1 in cholesterol biosynthesis [25], thus suggesting that At2g33630 could also be involved in plant sterol biosynthesis.

Alternatively, the lack of growth defects of the reported *hsd1 hsd2* double mutant could be caused by a weak impact of the T-DNA insertion on the *HSD2* gene. The reported *hsd2* mutant carried a T-DNA insertion in the seventh intron of the annotated *HSD2* gene and no data were shown in that study on the impact of the intron-localized T-DNA on the *HSD2* transcript [19]. It is quite possible that such an intron-inserted T-DNA could be spliced out by the Arabidopsis splicing machinery, which was known to occur in approximately 4% of analyzed intronic T-DNA inserts [26]. In this study, we used the CRISPR/Cas9-based genome editing technology to generate mutations in the two *HSD* genes with an A nucleotide insertion in the *HSD1* gene and a 46-nucleotide deletion in the *HSD2* gene. Both mutations caused a frameshift in the open-reading frame of the resulting *hsd1/hsd2* transcripts and consequential early translational termination (Figure 1b,c), thus likely being null mutations. Surprisingly, despite screening hundreds of T3/T4 offspring and identifying many *hsd1/+ hsd2* and *hsd1 hsd2/+* heterozygous double mutants, we were unable to obtain a single *hsd1 hsd2* double homozygous mutant. As expected, no detectable phenotypic change was seen in the *hsd1* and *hsd2* single mutant or the two heterozygous double mutants (Figure 2).

In recent years, the analysis of sterol biosynthesis defective mutants has revealed the functional significance of sterols in many plant developmental processes. The loss-of-function mutants of Arabidopsis sterol methyltransferase 1 (SMT1), which catalyzes the first S-adenosylmethionine-dependent C24 alkylation [27], exhibited pleiotropic defects, including shorter petioles, smaller and rounder leaves, delayed silique development, as well as abnormal embryonic development [27]. The Arabidopsis mutant defective in the cyclopropylsterol isomerase (CPI1), a plant-specific sterol biosynthetic enzyme converting the pentacyclic cyclopropyl sterols to tetracyclic sterols, was a dwarf mutant with small rounder and dark-green leaves, short hypocotyl, stunted roots, and defective gravity response [28]. Other sterol mutants, such as *cyp51a2* (sterol C14-demethylase), *fackel*/*hydra2* (sterol C14 reductase), and *hydra1* (sterol Δ^8^-Δ^7^ isomerase), all had defects in post-embryogenesis growth, exhibiting poorly developed hypocotyls, fused cotyledons, shortened roots, and seedling lethality [29,30,31,32]. Mutations in the enzymes involved in the downstream steps of the plant sterol biosynthetic pathway often cause phenotypes similar to those observed in mutants defective in BR biosynthesis, including dwarfism and reduced fertility, which could be rescued by exogenous BR applications. Our failure to obtain a homozygous *hsd1 hsd2* double mutant impeded our exploration of the significance of the two Arabidopsis 3βHSD/D enzymes in post-embryogenesis, as well as the functional redundancy of the two HSDs and other 3βHSD-like enzymes in sterol biosynthesis in vegetative tissues.

### 3.2. HSD1 and HSD2 Function Redundantly in Pollen Development

Our unsuccessful attempts to obtain the *hsd1 hsd2* double homozygous mutant, results of the reciprocal genetic crosses of the *hsd1/+ hsd2* and *hsd1 hsd2/+* heterozygous double mutants with their wild-type control, and our findings of both viable and non-viable pollen grains in the pollen sacs of both mutants suggest that HSD1 and HSD2 are critical for normal pollen development. Previous studies suggested that only those mutants defective in the initial steps of sterol biosynthesis showed an abnormal male gametophyte phenotype. For example, the simultaneous elimination of the two Arabidopsis genes encoding 3-hydroxy-3-methylglutaryl coenzyme A reductase (HMGR), which catalyzes the rate-limiting reaction in the mevalonate pathway and produces precursors for sterol biosynthesis, was shown to cause defective transmission of male gametes [33,34]. Similarly, loss-of-function mutations in the Arabidopsis cycloartenol synthase 1 (CAS1) that initiate the sterol biosynthesis by catalyzing the conversion of the linear 2,3-oxidosqualene molecule to the pentacyclic cycloartenol also cause defective transmission of male gametes [35]. The Arabidopsis genome encodes a single active squalene synthase (SQS) [36] and six homologs of the yeast squalene epoxidase (SQE) with three of them capable of rescuing a yeast SQE mutant [37]. Although there is no report on a loss-of-function mutant of a plant SQS, treatment of Arabidopsis plants with the well-studied SQS inhibitor squalestatin, which could also inhibit plant SQS [38], greatly reduced the plant fertility [33]. Interestingly, despite the presence of six SQE homologs in Arabidopsis, loss-of-function mutations in the Arabidopsis SQE1 gave rise to pleiotropic defects of plant growth and development, including infertility [37,39]. Despite all these reports suggesting the importance of sterol biosynthetic enzymes upstream of the cyclization reaction in male fertility, little is known about the underlying molecular mechanism(s) that link sterol biosynthesis to male gametophytic functions in plants. It is important to note that our study is the first that implicates the role of a post-cyclization sterol biosynthetic enzyme in male gametogenesis. Earlier studies on SMO1, which acts immediately upstream of the 3βHSD/D enzyme to remove the first C4 methyl group, did not detect any defect in male fertility but observed severe embryogenesis and post-embryogenesis defects [16]. However, this could be caused by incomplete elimination of all three SMO1-encoding Arabidopsis genes (*SMO1-1*, *SMO1-2*, and *SMO1-3*) because the reported study only examined the growth and developmental phenotypes of *smo1-1 smo1-2* and *smo1-1 smo1-3* double mutants [16]. It is also interesting to mention that mutations of the Arabidopsis SMT1, which catalyzes the initial C24 methylation before the first C4 demethylation, also lead to an embryogenesis defect with no apparent impact on male fertility [27]. This could also be caused by functional redundancy between SMT1 and its two homologs, SMT2 and SMT3, as all three enzymes were capable of catalyzing the first and second methyl addition at C24 when expressed in yeast and/or bacterial cells [27,40,41]. It will be interesting to determine if the complete elimination of SMTs or SMO1s could also cause male gametophytic lethality. More importantly, further studies are needed to know the biochemical and cellular causes of the pollen lethality of the *hsd1 hsd2* double mutation.

### 3.3. HSD1 and HSD2 Are Essential for Embryonic Development

To overcome the male fertility problem of the *hsd1 hsd2* double mutation and to determine the role of the two HSDs in vegetative growth, we constructed the *pLAT52::HSD2-FLAG* transgene with pollen-specific promoter *pLAT52*. Although such a transgene was able to fully rescue the pollen defect of the *hsd1 hsd2* double mutation, we did not obtain a transgenic plant containing *hsd1 hsd2* double mutation, suggesting the *hsd1 hsd2* double mutation could lead to embryo lethality. Thus, these two 3βHSD/D enzymes join other sterol biosynthetic enzymes in regulating embryogenesis, which include SMT1, SMO1/2, CYP51 (C14 demethylase), Fackel/Hydra1 (C14 reductase), and Hydra2 (Δ^8^-Δ^7^ isomerase) [15,16,27,29,30,31]. Together, these studies suggest a common mechanism by which mutations of these sterol biosynthetic enzymes affect embryonic development. One potential mechanism could be due to the significant reduction in the abundance of downstream sterols, such as campesterol and sitosterol. However, in Arabidopsis mutants defective in the downstream enzymes of the phytosterol biosynthesis pathway, such as *dwarf7* (*dwf7*, defective in Δ^7^-sterol C5 desaturase), *dwarf5* (*dwf5*, defective in Δ^5,7^-sterol C7 reductase), and *dwarf1* (*dwf1*, defective in Δ^24^-sterol Δ^24^ reductase), the embryogenesis process seems normal in these mutants despite the similar abundance reduction in the downstream sterols [42,43,44]. It is generally believed that the growth and developmental defects of the late-stage sterol-deficient mutants are mainly caused by the decreased abundance of BRs because most of their growth defects could be rescued by BR treatment [42,43,44]. By contrast, the exogenous application of BR was not able to rescue the embryonic developmental phenotypes of mutants defective in the early stage of the sterol biosynthetic pathway, such as *smt1*, *smo1-1 smo1-2*, *cyp51a2-3*, and *fackel* [16,27,29,30]. Our finding, coupled with those published studies, suggested that the reduced production of downstream sterols is unlikely the main cause of embryonic defects. We think that the embryogenesis defect might be attributed to the accumulation of certain sterol biosynthetic intermediates, which could be incorporated into the cellular membranes or function as signaling molecules, thereby affecting embryogenesis [30,32,45]. For example, the previously reported *smo1-1 smo1-2* and *smo2-1 smo2-2* mutants accumulate large amounts of 4,4-dimethylsterols and 4α-methylsterols, respectively [15,16], while the *cyp51a2-3* mutant accumulates 14α-methylsterols [29]. These sterol intermediates could somehow affect the biosynthesis or transport of auxin [28,31,46,47,48], which is known to be essential for normal embryogenesis [15,16]. Further experiments are needed to determine the abundance of various sterol intermediates in the relevant tissues of *pLAT52::HSD2-FLAG hsd1 hsd2/+* transgenic plants and to examine if treatment with auxin or other plant hormones could rescue some of the observed embryogenesis defects of the transgenic *hsd1 hsd2/+* mutants. It is also necessary to generate an *HSD2* transgene capable of rescuing both the male fertility defect and the embryo lethality of the *hsd1 hsd2* double mutation to determine whether or not the two HSDs also play important roles in the plant vegetative growth and development. Obtaining such a transgenic *hsd1 hsd2* double mutant will also indicate whether these two HSDs might function redundantly with the other two Arabidopsis homologs of the mammalian 3βHSDs, At2g43420 (RTN20) and At2g33630, in regulating certain aspects of plant vegetative development and stress tolerance.

## 4. Materials and Methods

### 4.1. Plant Materials and Growth Conditions

All Arabidopsis mutants and transgenic lines used in this research were in the Columbia-0 (Col-0) ecotype. The *hsd1*, *hsd2*, *hsd1/+ hsd2,* and *hsd1 hsd2/+* mutants were generated by the CRISPR/Cas9-based gene editing approach [49]. Stable transgenic plants were selected on half-strength Murashige and Skoog (½ MS) medium supplemented with 50 mg/mL hygromycin. Methods for seed sterilization and growth schemes for young seedlings and mature plants were described previously [50].

### 4.2. CRISPR/Cas9-Mediated Genome Editing to Create HSD1 and HSD2 Mutant

The CRISPR/Cas9 vectors used to create *hsd1/hsd2* mutants were provided by Dr. Qi-Jun Chen [49]. The target sites for introducing mutations into the *HSD1/2* genes while minimizing off-target mutations were selected using the web program CRISPR-GE (http://skl.scau.edu.cn/ (accessed on 16 November 2020)) [51] and the oligonucleotides for generating corresponding guide RNAs are listed in Appendix A. The CRISPR/Cas9 transgenes for introducing mutations in *HSD1/2* genes were constructed following a previously reported procedure [52]. The single-guide RNA (sgRNA) expression cassettes containing each target sequence were generated by overlapping PCR and cloned into the *pHEE401E* binary vector (https://www.addgene.org/71287/ (accessed on 16 November 2020)) using the Golden Gate cloning method [53]. The resulting construct was verified by PCR amplification, restriction enzyme digestion, and DNA sequencing. The *CRISPR/Cas9-HSD1/2* construct generated was introduced into the *Agrobacterium* strain *GV3101* by electroporation, which was subsequently used to transform Arabidopsis wild-type plants by the floral-dipping method [54]. T0 seeds were harvested, sterilized, and screened on ½ MS agar medium containing 50 mg/mL hygromycin for T1 lines carrying the *CRISPR/Cas9-HSD1/2* constructs. The resulting transgenic T1 lines were subsequently screened for intended mutations of *HSD1/2* genes by PCR amplification and subsequent DNA sequencing of the short genomic DNA fragments containing the intended mutation sites using gene-specific primers *HSD1-For/Rev* and *HSD2-For/Rev* (Appendix A). The DSDecodeM program (https://skl.scau.edu.cn/dsdecode/ (accessed on 18 February 2021)) of the CRISPR-GE tool kit was used for decoding the edited genomic sequence of targeted sites [51,55]. Segregated T2/T3 plants carrying no CRISPR/Cas9 transgene but homozygous/heterozygous for the confirmed mutations were identified by PCR analysis using the gene-specific primers and the *Cas9-For/Rev* primers (see Appendix A for their DNA sequences). The *HSD1-For/HSD1-Rev* primer set-amplified *HSD1* genomic fragment was digested by the restriction enzyme AvaII (New England Biolabs, Ipswich, MA, USA) that cuts the wild-type *HSD1* fragment but not the corresponding PCR fragment from the *hsd1* mutant carrying the A insertion. The *HSD2-For/HSD2-Rev* primers were used to amplify the *HSD2* genomic fragment. Because the *hsd2* mutation is caused by a 46-bp deletion, the *HSD2* PCR fragment from the wild-type allele is larger than the corresponding PCR fragment from the *hsd2* mutant allele, and simple electrophoresis of the *HSD2* PCR fragments could easily genotype *hsd2/hsd2*, *hsd2/HSD2*, and *HSD2/HSD2* plants.

### 4.3. Total RNA Extraction and cDNA Synthesis

Fifty milligrams of wild-type (Col-0) Arabidopsis seedlings were ground into a fine powder with liquid nitrogen. Total RNAs were extracted using the FastPure^®^ Plant Total RNA Isolation Kit (Vazyme, Nanjing, China) by following the manufacturer’s recommended protocol. After removing residual genomic DNAs, the extracted total RNAs were converted into the first-strand cDNA preparation using the HiScript^®^ III 1st Strand cDNA Synthesis Kit (+gDNA wiper) (Vazyme, Nanjing, China) and the manufacturer’s provided protocol. The resulting cDNA products were used immediately for PCR reactions or aliquoted and stored at −80 °C freezer.

### 4.4. Plasmid Construction and Plant Transformation

When constructing the *pC1305-pLAT52::HSD2-FLAG* vector, a 20-bp homologous arm sequence was added to the primer to match the vector, and the *HSD2* sequence was amplified using the wild-type Arabidopsis cDNA preparation as the template. A target DNA fragment was homologously recombined with the enzyme-linearized vector using homologous recombination enzymes. The *pC1305-pLAT52::MCS-FLAG* vector for generating the *pLAT52::HSD2-FLAG* transgene was kindly provided by Professor Jirong Huang [56]. To transform the plants, the plasmids were first introduced into the *Agrobacterium* strain *GV3101*, which was used to transform the plants via the floral dip method. All primers used for generating the transgene are listed in Appendix A.

### 4.5. Mutant Self-Inbred Offspring Separation Ratio and Forward and Reverse Cross Experiments

After harvesting individual heterozygous mutant seeds, they were germinated, and the resulting seedlings were used for genomic DNA extraction. The genotypes were verified by PCR, and the segregation ratio of the mutant in the self-pollinated progeny was calculated. Heterozygous mutants identified by PCR were used as pollen donors to pollinate wild-type stigma, and vice versa, to ensure maximum pollen transfer. The hybrid seeds were collected when mature and evenly sown onto ½ MS medium. The resulting young seedlings were harvested to extract genomic DNAs for PCR-based genotyping, which were subsequently used to calculate the transmission efficiency (TE) of male and female gametes. The TE was calculated by dividing the numbers of the heterozygous mutant with the total number of wild-type plants × 100% [57].

### 4.6. Flower Organ Phenotypic Observation and Pollen Alexander Staining

Mutant and wild-type plants of the same growth stage were selected. Flowers at the flat stage were sampled, and their anther morphology was observed and photographed by a volumetric microscope (Nikon: SMZ18; Nikon, Tokyo, Japan). Alexander staining is a simple method for assessing pollen viability, where viable pollen stains purple-red, while non-viable pollen remains green or pale grey due to the inability to take up the stain [21]. For flowers at the anthesis stage, Alexander’s staining solution was dropped onto a microscope slide, and the anther was placed directly into the staining solution to stain the pollen. Then, the Alexander staining solution was added to one side of the anther, covered with a coverslip, and gently pressed until the staining solution permeated the anther evenly. The sample was left at room temperature in the dark for ten minutes, and then observed and pictures were taken under a light microscope.

### 4.7. Phenotypic Observation of Pods and Statistical Analysis of Seed Abortion Rate

Mature green siliques were dissected under a dissecting microscope to reveal the developing seeds inside. The seeds inside the siliques were observed and photographed using a Nikon stereo microscope SMZ18 [58]. Approximately ten siliques at similar positions on the main inflorescent stem of mature Arabidopsis plants of different genotypes were collected. The siliques were opened and the number of normal and aborted seeds in each silique were recorded. The experiment was repeated for at least three times. The seed abortion rate of both wild-type and mutants was calculated and analyzed.

## Figures and Tables

**Figure 1 ijms-24-15565-f001:**
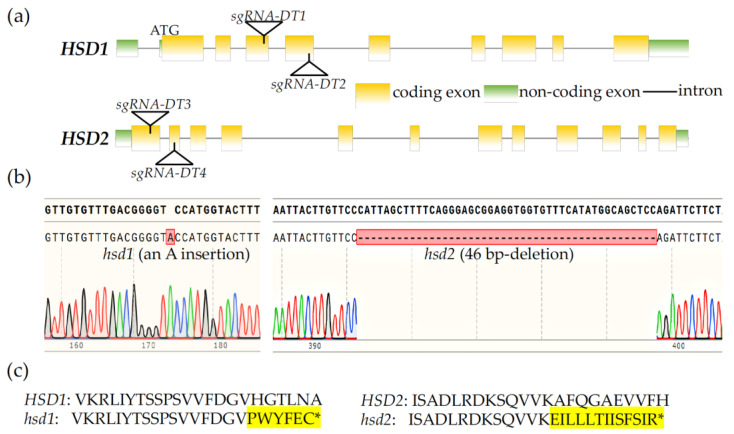
Generation of *hsd1 hsd2* mutants by the CRISPR/Cas9 genome editing approach. (**a**) Genomic location of guide DNA sequences (*sgRNA-DT1*, *sgRNA-DT2*, *sgRNA-DT3*, and *sgRNA-DT4*) targeting *HSD1* and *HSD2*. (**b**) Sequencing chromatograms of the CRISPR/Cas9-mediated genome editing of the *HSD1* and *HSD2* genes. (**c**) Amino acid sequences of the *hsd1* and *hsd2* mutants near the detected *hsd1* and *hsd2* mutation sites, respectively. * denotes the premature stop codon due to frameshift of the predicted mutant transcripts of *HSD1* and *HSD2*, while the highlighted amino acids are predicted extra amino acids from the mutant *HSD1*/*HSD2* transcripts.

**Figure 2 ijms-24-15565-f002:**
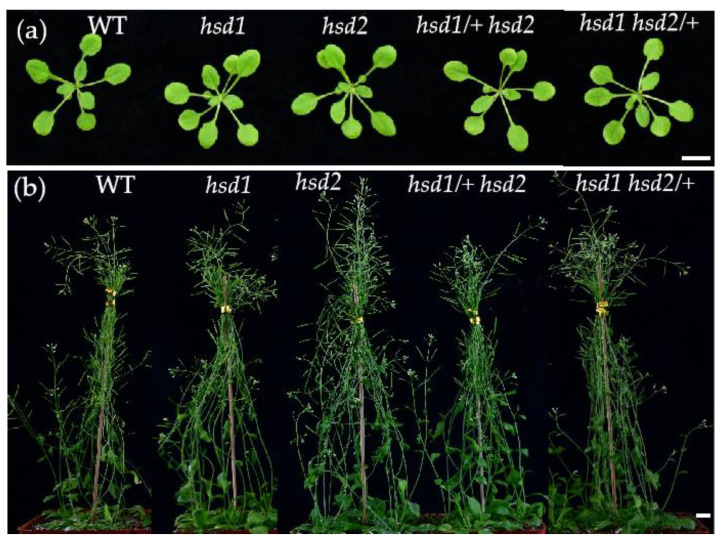
Phenotypic comparison of the single and heterozygous mutants of *HSD1* and *HSD2* genes and their wild-type control. (**a**) Photographs of 14-day-old soil-grown Arabidopsis plants. (**b**) Photographs of 35-day-old soil-grown mature Arabidopsis plants. In (**a**,**b**), scale bar = 1 cm.

**Figure 3 ijms-24-15565-f003:**
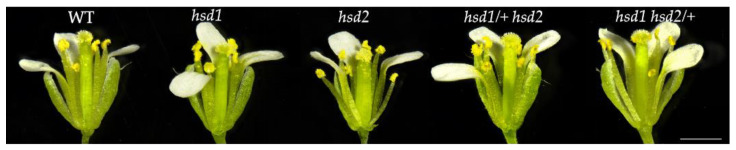
Phenotypes of flowers in the flattening phase of wild-type and mutants. Scale bar = 1 mm.

**Figure 4 ijms-24-15565-f004:**
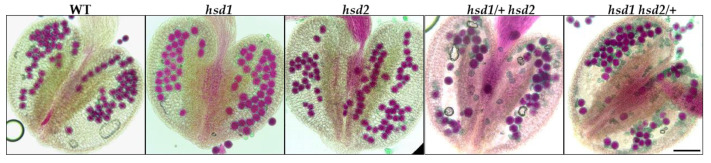
The *hsd1 hsd2* double mutation results in abnormal pollen development. Alexander’s staining of pollen grains in the anthers of wild-type Arabidopsis plants and indicated *hsd1/hsd2* mutants. Scale bar = 90 μm.

**Figure 5 ijms-24-15565-f005:**
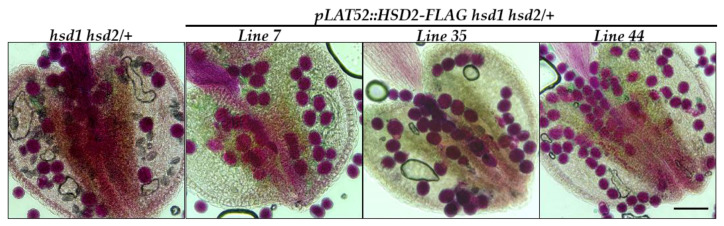
The *pLAT52::HSD2-FLAG* transgene rescued the pollen defect of the *hsd1 hsd2/+* mutant. Shown above are Alexander’s staining of pollen grains from several independent *pLAT52::HSD2-FLAG hsd1 hsd2/+* transgenic plants and their parental *hsd1 hsd2/+* mutant. Scale bar = 90 μm.

**Figure 6 ijms-24-15565-f006:**
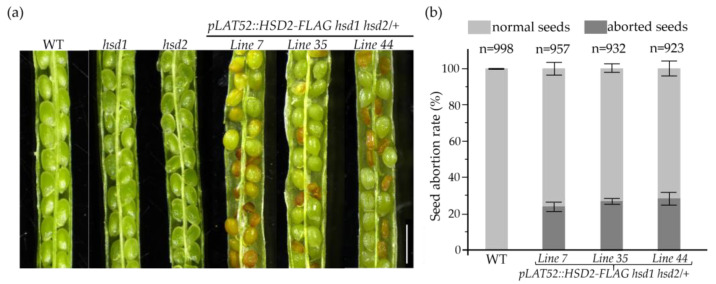
The *hsd1 hsd2* double mutation causes embryo lethality. (**a**) Photographs of opened mature siliques from the wild-type, two single *hsd* mutants, and three independent *pLAT52::HSD2-FLAG hsd1 hsd2/+* transgenic plants. Scale bar = 1 mm. (**b**) A bar graph presentation of the seed abortion rates of the three *pLAT52::HSD2-FLAG hsd1 hsd2/+* transgenic plants and their wild-type control. The numbers of seeds analyzed with standard errors are also shown on the graph.

**Table 1 ijms-24-15565-t001:** Genetic analysis of *hsd1/+ hsd2* and *hsd1 hsd2/+* mutants.

Self-Cross	No. of Progeny	Genotypes of Progeny		
		*HSD1 hsd2* or *hsd1 HSD2*	*hsd1/+ hsd2* or *hsd1 hsd2/*+	*hsd1 hsd2*
*hsd1/+ hsd2*	178	52.2% (93)	47.8% (85)	0
*hsd1 hsd2/+*	200	51.5% (103)	48.5% (97)	0

**Table 2 ijms-24-15565-t002:** Reciprocal cross-pollination of *hsd1/+ hsd2* and *hsd1 hsd2/+* mutants.

Male	×	Female	Expected Heterozygosity	Observed Heterozygosity	TE_M_	TF_F_
*hsd1*	×	WT	56/56	56/56	100%	NA
*hsd2*	×	WT	39/39	39/39	100%	NA
*hsd1/+ hsd2*	×	WT	120/240	0/240	0	NA
WT	×	*hsd1/+ hsd2*	118/235	120/235	NA	104%
*hsd1 hsd2/+*	×	WT	123/245	0/245	0	NA
WT	×	*hsd1 hsd2/+*	156/312	153/312	NA	96%

Expected values were based on the prediction that the mutant alleles were transmitted normally. TE_M_, male transmission efficiency; TE_F_, female transmission efficiency; NA, not applicable.

## Data Availability

The data presented in this study are available in this article and the Appendix A.

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
