# Peer review of "Arabidopsis 3β-Hydroxysteroid Dehydrogenases/C4-Decarboxylases Are Essential for the Pollen and Embryonic Development"

_ijms, 2023, doi:10.3390/ijms242115565_

Round 1
Reviewer 1 Report
"Arabidopsis 3β-Hydroxysteroid Dehydrogenases/C4-Decarbox-2 ylases are Essential for the Pollen and Embryonic Development" by Pan et al. is a study that presents high quality work.
The authors start very broad "Sterols are isoprenoid-derived lipids that have important physiological activities for all eukaryotic organisms" and "In the sterol biosynthesis pathways of plants, animals and fungi...", but I would have appreciated some more focussed discussion on the evolution of these specialised compounds in plants, see and cite e.g. Dadras, A., Rieseberg, T.P., Zegers, J.M.S. et al. Accessible versatility underpins the deep evolution of plant specialized metabolism. Phytochem Rev (2023). https://doi.org/10.1007/s11101-023-09863-2
And
Tim P. Rieseberg, Armin Dadras, Janine M.R. Fürst-Jansen, et al. Crossroads in the evolution of plant specialized metabolism, Seminars in Cell & Developmental Biology, Volume 134, 2023, Pages 37-58, https://doi.org/10.1016/j.semcdb.2022.03.004.
Further, I think a few more aspects could be discussed, along the questions of:
Are there indications of functional redundancy between HSD1, HSD2, and other related enzymes based on the sequence analysis?
Could you provide more details on the screening process for the T3/T4 offspring and the reasons behind the lack of phenotypic changes in the hsd1, hsd2 single mutants, and the heterozygous double mutants?
Were there any unexpected findings during the phenotypic analysis of these mutants?
How do the phenotypic outcomes of the hsd1 hsd2 double mutant compare to other sterol biosynthetic mutants, especially those affecting male fertility?
There are only few small mistakes. Overall, this is very good to follow.
Author Response
"Arabidopsis 3β-Hydroxysteroid Dehydrogenases/C4-Decarboxylases are Essential for the Pollen and Embryonic Development" by Pan et al. is a study that presents high quality work. The authors start very broad "Sterols are isoprenoid-derived lipids that have important physiological activities for all eukaryotic organisms" and "In the sterol biosynthesis pathways of plants, animals and fungi...", but I would have appreciated some more focussed discussion on the evolution of these specialised compounds in plants, see and cite e.g. Dadras, A., Rieseberg, T.P., Zegers, J.M.S. et al. Accessible versatility underpins the deep evolution of plant specialized metabolism. Phytochem Rev (2023). https://doi.org/10.1007/s11101-023-09863-2 and Tim P. Rieseberg, Armin Dadras, Janine M.R. Fürst-Jansen, et al. Crossroads in the evolution of plant specialized metabolism, Seminars in Cell & Developmental Biology, Volume 134, 2023, Pages 37-58, https://doi.org/10.1016/j.semcdb.2022.03.004.
Thanks for the comments. We have modified the opening sentence by adding “especially for plants that synthesize a wide variety of terpenoids including carotenoids, dolichols, tocopherols, chlorophylls, squalene [1].” The cited reference is the suggested publication 2. In the second paragraph of the introduction, we also speculate the significance of the different order of the three demethylation reactions in plants with the following sentence: The separation of the two C4-demethylation steps by the C14-demethylation might allow plants to produce unique 4-methylated sterols important for plant terrestrialization and plant stress tolerance [7, 11]. The reference 11 is the suggested publication 1.
Further, I think a few more aspects could be discussed, along the questions of:
Are there indications of functional redundancy between HSD1, HSD2, and other related
enzymes based on the sequence analysis?
Thanks for the suggestion. We think that HSD1 and HSD2 might be functionally redundant At2g43420 and At2g22630. We added the following sentence to the end of the discussion section: Obtaining such transgenic hsd1 hsd2 double mutant will also tell whether these two HSDs might function redundantly with the other two Arabidopsis homologs of the mammalian 3bHSDs, At2g43420 (RTN20) and At2g33630, in regulating certain aspects of plant vegetative development and stress tolerance.
Could you provide more details on the screening process for the T3/T4 offspring and the
reasons behind the lack of phenotypic changes in the hsd1, hsd2 single mutants, and the
heterozygous double mutants?
Thanks for asking. We added several sentences to the end of the 4.2 section of “Materials and Methods”. We also slightly modified the Usage for the HSD1-For/HSD1-Rev and HSD2-For/HSD2-Rev primer set in the Supplementary Table 2.
Were there any unexpected findings during the phenotypic analysis of these mutants?
Both the male sterility and the embryogenesis defect were unexpected findings. Given the earlier publication of the hsd1 hsd2 double mutant exhibiting no detectable growth/developmental defect, we thought that we will have to remove the other two of the Arabidopsis homologs of the mammalian 3bHSDs before we could detect a growth/developmental defect.
How do the phenotypic outcomes of the hsd1 hsd2 double mutant compare to other sterol biosynthetic mutants, especially those affecting male fertility?
Thanks for asking. We did discuss the phenotypes of the hsd1 hsd2 double mutant in comparison with reported mutants of the sterol biosynthesis, including the mutations that affect the MVA pathway, the early, middle and late steps of the phytosterol biosynthetic pathway.
Reviewer 2 Report
The manuscript "Arabidopsis 3β-Hydroxysteroid Dehydrogenases/C4-Decarboxylases are Essential for the Pollen and Embryonic Development" describes pollen and embryo viability of Arabidopsis 3βHSD/D knockout mutants obtained by CRISPR-Cas9 editing. Having observed altered allele segregation ratios, histological evidence, and genetic complementation, the Authors conclude that 3βHSD/D is required for pollen and embryo viability.
The article is well structured and presented, with clearly stated aims and an explicit justification for methodology choices. The results are clear and convincing. Only a few minor issues may be raised:
L166 - the heading does not seem correct. All the plants shown in fig. 3 have at least one functional allele of either hsd1 or hsd2, thus "double mutant" may be not appropriate. Suggest editing as follows: "hsd1 hsd2 double mutation in heterozygosity does not affect flower formation".
L181 - not sure if "gametophytic lethality" is appropriate in this context, since the Authors observe non-viable pollen rather than gametophyte. Suggest replacing with "pollen lethality"; other more generic forms (eg. gametophytic defect) may be acceptable.
Please check the attached file for reference. The article is acceptable for publication after these minor edits.

Author Response
The manuscript "Arabidopsis 3β-Hydroxysteroid Dehydrogenases/C4-Decarboxylases are Essential for the Pollen and Embryonic Development" describes pollen and embryo viability of Arabidopsis 3βHSD/D knockout mutants obtained by CRISPR-Cas9 editing. Having observed altered allele segregation ratios, histological evidence, and genetic complementation, the Authors conclude that 3βHSD/D is required for pollen and embryo viability.
The article is well structured and presented, with clearly stated aims and an explicit justification for methodology choices. The results are clear and convincing. Only a few minor issues may be raised:
Thanks for your support!
L166 - the heading does not seem correct. All the plants shown in fig. 3 have at least one functional allele of either hsd1 or hsd2, thus "double mutant" may be not appropriate. Suggest editing as follows: "hsd1 hsd2 double mutation in heterozygosity does not affect flower formation".
Thanks for spotting the mistake. We change the heading based on your suggestion.
L181 - not sure if "gametophytic lethality" is appropriate in this context, since the Authors observe non-viable pollen rather than gametophyte. Suggest replacing with "pollen lethality"; other more generic forms (eg. gametophytic defect) may be acceptable.
Following your suggestion, we changed “gametophytic lethality” to “pollen lethality”.
Please check the attached file for reference. The article is acceptable for publication after these minor edits.
Thank you very much for your carefully reading the manuscript and identified those marked errors. We made the suggested changes in the revised manuscript.